# Engagement in meaningful activities post suicide loss: A scoping review protocol

**Monique Gill**[1]*, **Meera Premnazeer**[1], **Orianna Scali**[1], **Sakina Rizvi**[3,4], **Alex Schendelman**[5], **Helene Polatajko**[1,2]�, **Jill I. Cameron**[1,2]☳

**1** Rehabilitation Sciences Institute, Temerty Faculty of Medicine, University of Toronto, Toronto, Ontario, Canada, **2** Department of Occupational Science & Occupational Therapy, Temerty Faculty of Medicine, University of Toronto, Toronto, Ontario, Canada, **3** Arthur Sommer Rotenberg Suicide and Depression Studies Program, St. Michael's Hospital, Toronto, Ontario, Canada, **4** Department of Psychiatry, Temerty Faculty of Medicine, University of Toronto, Toronto, Ontario, Canada, **5** Distress Centres of Greater Toronto, Toronto, Ontario, Canada

☳ These authors contributed equally to this work.
* moniquekaur.gill@mail.utoronto.ca

**Data Availability Statement:** No datasets were generated or analysed during the current study. All relevant data from this study will be made available upon study completion.

## Abstract

### Rationale

Each day, more than 10 Canadians die by suicide. Each suicide leaves entire communities to manage the traumatic aftermath of this loss. Individuals bereaved by suicide loss are at a higher risk of experiencing negative mental health outcomes. Current research suggests that engagement in meaningful activities may be an avenue to protecting mental health. It is important to understand if this is also the case for those experiencing bereavement post suicide loss. To date, there has not been a synthesis of the literature examining suicide loss and the nature and extent of engagement in meaningful activities post loss.

### Objectives

1) To describe the nature and extent of the peer-reviewed suicide loss and bereavement literature related to engagement in meaningful activities; and 2) to identify facilitators and barriers that may impact engagement in meaningful activities post loss.

### Methods

This paper describes a scoping review protocol that will be completed using stages identified by Arksey and O'Malley and updated by Levac and colleagues. Joanna Briggs Institute framework will also guide this review. Four electronic databases will be searched for suicide bereavement/loss concepts. Two reviewers will apply inclusion and exclusion criteria to identify articles discussing engagement in meaningful activities of everyday living post loss. Data will be descriptively summarized and analyzed using inductive content analysis. Results will be reported following PRISMA Extension for Scoping Reviews.

### Expected results

A descriptive summary and conceptual map describing the current state of the peer-reviewed literature will be constructed.

**Funding:** M. Gill, PhD Candidate at the University of Toronto, has been supported by the Ontario Graduate Scholarship, Toronto Rehabilitation Institute Student Scholarship, the Dawson Family Scholarship and the Peter Rappolt Family Scholarship for Research in Occupational Performance and Wellbeing in relation to this work. The funders did not and will not have a role in study design, data collection and analysis, decision to publish, or preparation of the manuscript.

**Competing interests:** The authors have declared that no competing interests exist.

## Conclusion

Experiencing a suicide loss increases the risk of negative mental health outcomes. A synthesis of literature is required to map the current available evidence related to suicide bereavement and engagement in meaningful activities, with potential implications for improving supports and services for those bereaved. This protocol is register with Open Science Framework Registries (10.17605/OSF.IO/M2NES).

## Introduction

Among Canadians, there has been a steady decline in mental health status since 2007 [1]. According to Statistics Canada, 77% of youth and young adults ages 15–30, 75% of adults ages 31–46, and 72% of middle aged to older adults ages 47 and older, reported having excellent to good mental health in 2007 [1]. This percentage has steadily declined to 60% among youth and young adults, 66% among adults, and 69% among middle aged to older adults in 2019 [1]. On average more than 12 Canadians die by suicide everyday [2]. Suicide remains the second leading cause of death for those between the ages of 15–34, only surpassed by accidental deaths [3]. Each suicide leaves behind family members, friends, peers, and entire communities to manage the traumatic aftermath of this sudden loss. Those left behind are often referred to as survivors of suicide loss, suicide survivors or individuals bereaved by suicide [4]. These individuals will be referred to as survivors of suicide loss throughout this scoping review protocol.

Each suicide death leaves behind approximately six individuals [5]. This estimate was not based on empirical evidence, however, has been echoed widely [6]. This estimate closely aligns with what has been reported through Statistics Canada with estimates of 7 to 10 individuals impacted by each suicide loss [2]. However, more recently a random-digit dial survey found that approximately 135 individuals are exposed to each suicide within the United States [6]. This large discrepancy questions this widely accepted estimate and further speaks to the importance of understanding the impact of loss on the lives of those exposed to suicide.

Although consensus among researchers has yet to be reached regarding the definition of survivors of suicide loss, recent studies are moving towards providing a common language to identify and categorize the relationship and self-reported degree of closeness to the deceased [7–10]. Bereavement experiences can vary drastically for all types of survivors of suicide loss [10]. Cerel and colleagues suggest a continuum of survivorship with the following four categories (from low to high degree of closeness): 1) suicide exposed (e.g., those who know of someone who died by suicide; however, did not personally experience psychological distress / consequences related to the loss or any long term impacts); 2) suicide affected (e.g., those who lost someone to suicide and experienced psychological distress); 3) suicide bereaved, short term (e.g., those who were close to / had an attachment relationship to the individual who died by suicide and experienced a short term response to bereavement); and 4) suicide bereaved, long term (e.g., those who were close to / had an attachment relationship to the individual who died by suicide and experienced a long term response to bereavement) [8]. The latter three describe situations in which the death has significantly impacted the lives of survivors leading to negative functional or health-related consequences whether temporary/short term (e.g., suicide affected and suicide bereaved short term) or long term and chronic (e.g., suicide bereaved long term). The planned scoping review will include individuals that fall into any of the categories described by Cerel and colleagues [8].

Research has yet to be completed on specific physical and mental health consequences as they relate to the categories outlined in the proposed continuum of survivorship [8]. However,

research conducted with survivors of suicide loss across the continuum suggests that they are at a higher risk of experiencing negative physical and mental health outcomes as compared to the general population [2, 4, 11, 12]. Pitman and colleagues compared survivors of suicide loss to those bereaving other types of loss [4]. They observed an increased risk of suicide, psychiatric admissions, and mental illness [4]. An exposure to suicide has also shown to increase the risk of post-traumatic stress disorder [13], depression [14], complicated grief [15], and suicidal ideations [16]. Further, Linde and colleagues studied the grief process following the experience of suicide loss, concluding that it is highly complex due to factors impacting those left behind [17]. In addition to the bereavement of an individual, survivors of suicide loss experience stigma, shame, guilt, need to conceal events surrounding death, blame, rejection, exposure to prior suicide attempts by deceased, familial history of mental health challenges, and strained relationships prior to and following the death [17]. These additional factors contribute to a complicated grief reaction or process, where individuals face further challenges adjusting to life post-loss [17, 18]. Complicated grief reactions are also associated with disruptions in daily functioning which can include a decrease in participation in meaningful activities [19, 20].

As described within occupational therapy literature, this scoping review will define meaningful activities as any task related to self-care, productivity, and leisure "that is performed with some consistency and regularity, that brings structure, and is given value and meaning by individuals and a culture" (i.e., occupation) [21, 22 p. 19]. The occupational therapy and occupational science literature suggests that engagement in meaningful activities can help individuals find meaning in life, ultimately, supporting recovery, meeting basic psychological needs and in turn, improving physical and mental health [23, 24].

There have been several previous reviews of the suicide bereavement and loss literature. A systematic mapping review of this literature completed by Maple and colleague [25], highlights that most studies since 1970 have focused on attitudes towards suicide, stigma, impact on health professionals, comparing types of loss, ethical papers on studying survivors, effectiveness of interventions, estimating exposure, and the experience of survivors, including cultural minorities/indigenous groups. More recent scoping reviews completed by Higgins and colleagues [26] and Kaspersen and colleagues [27], focused on peer-led interventions, and supports / services for those bereaved by suicide. Shields, Kavanagh and Russo completed a systematic review of the qualitative literature focusing on the bereavement process following suicide loss [28]. Their review highlighted three aspects of the bereavement process discussed often in qualitative literature: "the feelings following bereavement by suicide, the process of making meaning of the event, and the social context in which the feelings and meaning-making process occur" [28 p. 447]. This review also emphasized the important role that the meaning making process plays in adjusting to life post loss [28]. A search of the Open Science Framework database for registered scoping review protocols did not identify any literature reviews or protocols with the primary aim of exploring meaningful activities among individuals who have experienced a suicide loss. This protocol has been registered on Open Science Framework Registries (10.17605/OSF.IO/M2NES) and any amendments to this protocol will be reported in the completed scoping review.

Due to complexities related to the experience of suicide loss, the negative mental and physical health outcomes, and the increased risk of experiencing complicated grief, it is imperative that further research be undertaken. This research must focus on exploring concepts such as engagement in meaningful activities, that have shown potential in other contexts to act as protective factors to assist in bettering supports and services for those experiencing a loss. To date, a review synthesizing the literature as it relates to engagement in meaningful activities post suicide loss has yet to be completed. The objectives of this proposed scoping review are to: 1) explore the nature and extent to which the peer-reviewed, suicide loss and bereavement

literature addresses engagement in meaningful activities of everyday living; and 2) identify key facilitators and/or barriers that may impact engagement in meaningful activities post-loss.

## Methods

### Design

This scoping review will be guided by scoping review stages identified by Arksey and O'Malley [29] and updated by Levac and colleagues [30], while being guided by the methodology outlined by Joanna Briggs Institute (JBI) [31]. As recommended by JBI methodology, this review will also follow the Preferred Reporting Items for Systematic Reviews and Meta Analyses Extension for Scoping Reviews (PRISMA-ScR) [32]. This mixed methodological approach was chosen as authors were familiar with Arksey and O'Malley's review stages, however, recognized that updates from Levac and colleagues, as well as the JBI methodology are essential as they increase rigour and clarity [29–31]. Within each stage outlined below, Levac and colleagues methodology, as well as the JBI methodology is reviewed to ensure all steps are rigorous [30, 31]. For example, within stage 1, the JBI methodology suggests defining and aligning the objectives of the scoping review to the research questions which is completed below.

This scoping review will follow six stages: 1) identifying the research question; 2) identifying relevant studies; 3) study selection; 4) charting the data; 5) collating, summarizing, and reporting the results; and 6) consultation. This protocol will now go on to describe each stage. The literature search will be conducted in late January 2023 and the goal for manuscript submission will be set for March 2024 following an updated search.

### Stage 1: Identifying the research question

The planned review will answer the following primary research question: *What is the nature and extent to which the peer-reviewed, suicide loss and bereavement literature addresses engagement in meaningful activities of everyday living (i.e., occupation)*?

The secondary question for this planned review is: *What key facilitators and/or barriers impacting engagement in meaningful activities post loss are described within the literature*?

Addressing these questions will ultimately meet the objective of this scoping review which is to synthesize the peer-reviewed suicide bereavement and loss literature as it relates to engagement in meaningful activities through describing the nature and extent of this literature, as well as the key facilitators and/or barriers related to engagement in meaningful activity.

### Stage 2: Identifying relevant studies

The Population-Concept-Context (PCC) framework provided by the JBI for scoping reviews will be used to inform the search strategy and identify relevant studies [Table 1; 31].

**Inclusion criteria.** To be included studies must: 1) be in a published peer-reviewed journal article; 2) be considered an original literature review or a qualitative/quantitative/mixed methods study, written in English; 3) include participants impacted by suicide loss and

**Table 1. PCC framework: Inclusion and exclusion criteria.**

| | |
|---|---|
| Population | Individuals impacted by suicide falling any of the following categories: 1) suicide exposed; 2) suicide affected; 3) suicide bereaved, short term; and 4) suicide bereaved, long term [8]. Due to limited literature in the field, all age groups will be included. |
| Concept | Engagement in meaningful activities following a suicide loss and identification of possible facilitators and barriers to engagement post-loss |
| Context | Limited to suicide loss literature after 1969 to present day and limited to the English language |

bereavement (mixed sample studies are only to be included if they indicate independent results for suicide loss survivors); 4) discuss living/meaningful activity/activities of everyday living in any way and any capacity (e.g., as interventions, postventions, personal stories, quotes, etc.) within the study; and 5) published after 1969. Research on suicide postvention and post-loss intervention gained momentum in the late 1960's [33, 34]. During this time, terminology concerning the concept of suicide loss began to develop into what it is today [8, 33, 34]. Therefore, this scoping review will be limited to suicide loss literature after 1969. Additional studies will be identified by reviewing the reference lists of included articles and previous review articles following the full text review.

**Exclusion criteria.** This review will not include grey literature, books, book chapters, unpublished studies, dissertations, abstracts, and reports. In addition to barriers such as time and resource constraints, these criteria will enable the exclusion of non-peer reviewed work which will assist in ensuring studies included within the review will be of better methodological quality.

**Search strategy.** Following multiple consultations with two rehabilitation sciences librarians working at the University of Toronto, two major concepts were identified to assist in the creation of the search strategy: 1) suicide; and 2) bereavement/loss. Meaningful activities of daily living or activities of daily living were not included as, during an exploratory search (completed December 2022), these concepts drastically narrowed the search results and omitted many studies that addressed specific activities within their title/abstract or study aim(s)/ question(s)/objective(s) that may fall within the definition of meaningful activities of everyday living listed in the introduction.

The JBI guiding framework outlines three steps to creating a comprehensive search strategy [31]. These steps will allow the research team to ensure that the concepts of suicide and bereavement/loss are being explored to their full capacity to identify articles discussing meaningful activities. First, an initial limited search will be completed using MEDLINE and PsychInfo. The following search strategy will be used:

1. exp suicide (Medical Subject Heading (MeSH))

2. suicid*

3. 1 or 2

4. exp bereavement (MeSH)

5. (bereav* OR loss OR grie* OR mourn*)

6. 4 or 5

7. 3 and 6

During this step, an analysis of the first 10 retrieved articles from each database will be done examining the text within the title, abstract, and identified index terms to ensure all appropriate search terms are included within the search strategy. Next, a second search with all previously and newly identified keywords and index terms, as well as any newly identified keywords and index terms, will be done across all included databases. Again, the first 10 retrieved articles will be examined, and the search strategy will be updated.

To increase comprehensiveness, a novelle cascaded search method will be used to identify any missing or relevant terms to further improve this search strategy. The cascaded search method employs a year-by-year search of the literature to determine if additional relevant text words or index terms should be added. Articles published within a single year will be examined to determine if any alterations can be made to assist in increasing the number of relevant

results. All relevant text words within the title and abstract, as well as index terms, used to describe the first 10 articles will be added to the search strategy. The cascaded search method is an iterative process, examining the search strategy across several calendar years. This is completed until new text words or index terms no longer emerge. Finally, JBI methodology outlines the completion of a reference list search following the full text review as the final step in the creation of a comprehensive search strategy.

An in-depth example of a search strategy for MEDLINE can be found in S1 Appendix. Relevant databases chosen in collaboration with librarians at the University of Toronto include MEDLINE, EMBASE, PsychInfo and CINAHL. The methods described above will allow for an extensive search of these four databases.

## Stage 3: Study selection

All identified articles will be exported to Covidence, a screening and data extraction tool that supports literature reviews [35], to remove duplicates and facilitate the screening process. Inclusion criteria will be pilot tested prior to the title/abstract review in order to determine whether: 1) inclusion/exclusion criteria are expressed clearly; and 2) reviewers are interpreting the eligibility criteria appropriately/consistently. The first 25 articles to appear on Covidence will be selected [31]. All reviewers will complete a blinded pilot test to achieve 80% agreement [31]. Reviewers will then consult the research team to discuss conflicts or possible areas of improvement of the study selection process. Two independent reviewers will screen each article through the title and abstract review, and full text review. Reviewers will not be the same for each article and multiple reviewers will be involved throughout the process. The review team consists of health researchers well-versed in the completion of literature reviews [36–38].

**Title and abstract review.**   Each title and abstract will be independently reviewed by two members of the research team using the inclusion and exclusion criteria. Conflicts regarding inclusion of articles will be managed by a third reviewer. Studies deemed to meet inclusion criteria will move onto a full text review.

**Full text review.**   The full text review will be completed by two reviewers. Again, conflicts will be managed by a third reviewer. Covidence automatically generates a PRISMA-ScR flow diagram, reporting decisions the research team made when assessing articles for inclusion.

**Quality appraisal.**   The quality of included studies will be reported using the appropriate critical appraisal tool as provided by JBI to assist in determining the trustworthiness and rigour of the results [39]. For example, if an included study uses a qualitative design, the checklist for qualitative design referenced by the JBI will be used to critically appraise the quality of the study [39]. This scoping review will include a wide range of study designs and methodologies addressing a broad range of research questions with the intention being to identify the full nature and extent of the discussion on engagement in meaningful activities post lost. There is no intent to exclude articles based on the results of the critical appraisal as this review is not evaluating the effectiveness of such engagement.

## Stage 4: Charting the data

Two reviewers will extract descriptive data using an excel table. For all studies the following descriptive information (i.e., data items) will be collected: 1) title of the study; 2) author(s); 3) name of journal; 4) publication year and country; 5) specific study design; and 6) sample description (e.g., age, sex, gender, time since loss, and nature of relationship with individual lost) and size. Through collection of this descriptive information, the nature (e.g., methodologies, sample descriptions/size, etc.) and extent (e.g., number of studies) of the peer-reviewed literature will be explored. This method of charting data will be pilot tested by two reviewers

with three studies [30]. This process will contribute to consistency among data extractors and assist with refining the data extraction process. Charting will remain an iterative process, where additional unforeseen, useful descriptive data may be added.

## Stage 5: Collating, summarizing, and reporting the results

Inductive descriptive qualitative content analysis will be undertaken to further examine the nature and extent (e.g., discussion of life/living areas, daily activities, meaningful activities, facilitators or barriers discussed related to engagement in activity, outcomes/results of the studies, etc.) of meaningful activities of everyday living [40]. All full text of articles will be imported into NVivo v12, a data analysis software [41], for the completion of content analysis. Steps outlined by Elo and Kyngäs for inductive content analysis of qualitative data will be followed [40].

Inductive descriptive qualitative content analysis was chosen to generate an understanding of what is currently known about the experience of suicide loss in relation to meaningful activities. By completing an inductive qualitative content analysis, more information will be gathered about the topic (e.g., nature and context of meaningful activities post suicide-loss). This method is especially useful and applicable to topics that are lacking in knowledge or where knowledge may be fragmented such as research about meaningful activity and suicide loss [40]. The goal of this method is data driven—to find patterns within qualitative data from several sources and combine these codes into higher levels of abstraction described through categories or themes [40, 42]. This process will assist in providing fulsome answers to the primary and sub-questions of this scoping review.

This inductive content analysis will be completed independently by two reviewers. Phase one, or the preparation phase, will involve becoming familiar with the data through several readings of each article to determine: 1) aim(s)/ objective(s)/question(s) of the study; 2) life/living areas, daily activities, meaningful activities discussed; 3) facilitators or barriers discussed related to engagement in activity, if any; and 4) outcomes/results of the study. These components of each article will be highlighted within Nvivo v12 [41]. Once familiar with the data, the two reviewers will select a unit of analysis. The unit of analysis is dependent on the content of the data and its relation to the primary and secondary questions. This could include a particular section of identified articles (e.g., the results and discussion sections) or the full text.

Phase two, the organizing phase, will include open coding in which notes, and headings (i.e., general themes) will be attributed to sections of the text independently by each reviewer. During this phase, the reviewers will reflect on the primary and secondary questions posed by this review and construct notes / headings related to the following: 1) life/living areas, daily activities, and meaningful activities discussed; 2) facilitators or barriers discussed related to engagement in activity, if any; and 3) outcomes/results of each study. The text is read over multiple times until no new notes or headings are constructed. Reviewers will work together to compare collected notes and headings to freely create sub-categories. These sub-categories will then be grouped into higher order categories known as generic categories where the aim is to understand the unit of analysis (e.g., results/discussion sections or full text of selected studies) as opposed to simply categorizing based on similarity. Abstraction will then be completed in which generic categories will be grouped into main categories. All levels of categories will be reported using a conceptual map included in the results section of this scoping review.

**Presentation of results.** Data regarding the search and inclusion of articles will be reported in line with the PRISMA-ScR reporting guidelines, including a PRISMA-ScR flow diagram. This scoping review will include a descriptive summary that will be presented in a tabular format with the following headings: 1) title of the study; 2) author(s); 3) name of journal; 4)

publication year; 5) specific study design; and 6) sample description and size. Results of the inductive content analysis will also be included in the form of a written summary in conjunction with a conceptual map. Following this, reported results will be linked back to the aims of this scoping review.

**Stage 6: Consultation.** During the previous stage, consultation will occur through a workshop with the research team (consisting of occupational therapy/suicide/health researchers and community partners from the Distress Centres of Greater Toronto) to discuss the findings of this scoping review. This consultation will assist in clarifying best methods for interpreting, reporting, and disseminating results, as well as allow for the opportunity to discuss the congruency, or lack thereof, seen between practice and research.

## Limitations

This review does present with some limitations. To begin, as there has been a limited focus directly on meaningful activities within the suicide bereavement and loss literature, inclusion and exclusion criteria may not be easily applied to all studies leading to a potentially time-consuming review or the exclusion of relevant articles. This may be caused by articles not reporting the full scope of their results within their abstracts (e.g., not mentioning results related to adjustment to life, activities of daily living or meaningful activities within their abstracts) or abstracts within bereavement and loss literature not reporting suicide loss survivors as participants. This will be addressed through frequent consultation with the research team to allow for continuous clarification and refining of inclusion and exclusion criteria as reviewers become more familiar with language used within this field through the abstract and title review. In addition, this scoping review will only include articles in English and will not include grey literature, books, book chapters, unpublished studies, dissertations, and reports. This may also exclude relevant articles published in other languages and may miss relevant information found within other types of evidence.

Finally, the quality of included studies will be appraised; however, studies will not be excluded based on the results of their appraisal. This allows for the potential inclusion of lower quality articles. This decision was made as the scope of this review is large, including a wide range of methodologies and study designs. The intention of this review is to identify the nature and extent of the discussion on engagement in meaningful activities of everyday living post lost. This review is not aiming to evaluate the effectiveness of such engagement.

## Supporting information

**S1 Appendix. This is the search strategy for MEDLINE via Ovid.**
(DOCX)

**S2 Appendix. This is the PRISMA-P 2015 checklist.**
(DOCX)

## Acknowledgments

We would like to thank and acknowledge the contributions of Erica Nekolaichuk and Julia Martyniuk, Liaison & Education Librarians at the Gerstein Science Information Centre–University of Toronto, for providing guidance on search strategy development, and the completion of a comprehensive literature search.

## Author Contributions

**Conceptualization:** Monique Gill, Helene Polatajko, Jill I. Cameron.

**Data curation:** Monique Gill, Meera Premnazeer, Orianna Scali, Helene Polatajko, Jill I. Cameron.

**Funding acquisition:** Monique Gill.

**Methodology:** Monique Gill, Meera Premnazeer, Helene Polatajko, Jill I. Cameron.

**Project administration:** Monique Gill.

**Supervision:** Sakina Rizvi, Helene Polatajko, Jill I. Cameron.

**Writing – original draft:** Monique Gill.

**Writing – review & editing:** Monique Gill, Meera Premnazeer, Orianna Scali, Sakina Rizvi, Alex Schendelman, Helene Polatajko, Jill I. Cameron.

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
