## [Decision Letter · Decision Letter 0]

9 Nov 2023

PONE-D-23-25565Engagement in meaningful activities post suicide loss: A scoping review protocol

PLOS ONE

Dear Dr. Gill,

Thank you for submitting your manuscript to PLOS ONE. After careful consideration, we feel that it has merit but does not fully meet PLOS ONE’s publication criteria as it currently stands. Therefore, we invite you to submit a revised version of the manuscript that addresses the points raised during the review process.

We look forward to receiving your revised manuscript.

Kind regards,

Grant Rich, Ph.D.

Academic Editor

PLOS ONE

2. Please amend either the abstract on the online submission form (via Edit Submission) or the abstract in the manuscript so that they are identical.

3. We notice that your supplementary file is included in the manuscript file. Please remove them and upload them with the file type 'Supporting Information'. Please ensure that each Supporting Information file has a legend listed in the manuscript after the references list.

Additional Editor Comments:

Dear Author, two reviewers and I carefully reviewed your submission and the decision is "MINOR REVISION"

Please attend to the reviewer comments below when making your revision,

Warmly, Dr Rich

Juneau, Alaska

BOTH REVIEWERS VOTE MINOR REVISION

Engagement in meaningful activities post suicide loss: A scoping review protocol"

REVIEWER one minor revision

The proposal proposes to review studies (published after 1969) on post-loss engagement in meaningful activities by those people who were affected by suicide. The proposal has been pre-registered on Open Science Framework Registries. The proposal follows relevant guidelines for conducting scoping reviews. I recommend its publication if the journal has a policy to publish study proposals rather than completed studies. I do have some concerns with the statement of research questions on page 6:

1) What is the nature and extent to which the peer-reviewed, suicide loss and bereavement literature addresses engagement in meaningful activities of everyday living (i.e., occupation)?

2) What key facilitators and/or barriers impacting engagement in meaningful activities post-loss are described within the literature?

The above two broadly stated research questions do not address aspects stated in the abstract (Conclusions Part):

“Experiencing a suicide loss increases the risk of negative mental health outcomes. However, engagement in meaningful activities may improve mental health outcomes and have implications for improving supports and services for those bereaved.”

It seems the proposal needs to consider also reviewing—what type of mental health concerns are reported by people who have been exposed to and affected by suicide, and suicide bereaved short and long term (as stated on page 7). The question of whether exposure, etc. increases the risk of mental health issues is not addressed in the questions. This would require reviewing studies that specifically examine those studies assessed risks associated with exposure etc. Furthermore, it would be helpful to define these terms: “exposed to” “affected by” and “bereaved short” and long term.” It would also be helpful to define “meaningful activities” – are these specifically undertaken for coping purposes (e.g., seeing a therapist, joining a self-help group, reading self-help books on bereavement)? Can continuing to engage in daily living activities (e.g., brushing one’s teeth, taking showers, eating meals) be meaningful? It is not clear whether they plan to examine if they plan on reviewing whether engaging in meaningful activities indeed improves mental health outcomes. Will they be identifying the “meaningful” activities that studies have found to improve mental health outcomes? It would be helpful to have these points clarified in the proposal.

--

REVIEWER 2 minor revision

Overall, this is a solid manuscript, what's missing are the results and the interpretation, as well as some minor edits throughout the manuscript.

For the intro:

- be more specific, add in the stats and number detailing the decline of mental health status

- for the term "suicide survivor" - This label may be misconstrued (e.g., readers may interpret this as someone who survived a suicide attempt), add additional context or reframe.

- Re: Cerel et al reference (lines 33 through 35) - Add in specific reference for Canada. If there are limited studies or no studies, mention this. By doing so, it provides context as to why the Cerel et al reference was used.

- Re: Cerel et al reference (lines 37 - 39): Additional References? Supplement and support Cerel at al reference with additional references/studies.

- Awkward sentence: (Lines 50 - 54): Awkward sentence; rephrase this sentence - make it more concise.

For the Methods:

- Brief explanation as to why you chose the scoping review protocol by Arksey & O'Malley

- Brief explanation of whether or not the "two reviewers" are the same throughout this process or will they be rotated.

- Table 1  Population - Consider adding demographics and other details/information about the makeup of this population

- line 125 - grammar  ...began to develop [to] what it is today...  change to  "into"; rework this sentence, sounds awkward.

- Exclusion Criteria - Consider adding in a brief explanation as to why these are not being included.

- Stage 3 - study selection - Provide a brief explanation of how this will be randomized.

- Quality Appraisal - Provide 1-2 examples

For the Limitations

- When results are ready - describe the results, interpret the results, discuss the potential implications and significance of the study.

- Discuss/identify next steps. Add in discussion/conclusion

Reviewers' comments:

Reviewer's Responses to Questions

**Comments to the Author**

1. Does the manuscript provide a valid rationale for the proposed study, with clearly identified and justified research questions?

Reviewer #1: Yes

Reviewer #2: Yes

2. Is the protocol technically sound and planned in a manner that will lead to a meaningful outcome and allow testing the stated hypotheses?

Reviewer #1: Partly

Reviewer #2: Partly

3. Is the methodology feasible and described in sufficient detail to allow the work to be replicable?

Reviewer #1: Yes

Reviewer #2: Yes

4. Have the authors described where all data underlying the findings will be made available when the study is complete?

Reviewer #1: No

Reviewer #2: Yes

5. Is the manuscript presented in an intelligible fashion and written in standard English?

Reviewer #1: Yes

Reviewer #2: Yes

6. Review Comments to the Author

You may also provide optional suggestions and comments to authors that they might find helpful in planning their study.

Reviewer #1: The proposal proposes to review studies (published after 1969) on post-loss engagement in meaningful activities by those people who were affected by suicide. The proposal has been pre-registered on Open Science Framework Registries. The proposal follows relevant guidelines for conducting scoping reviews. I recommend its publication if the journal has a policy to publish study proposals rather than completed studies. I do have some concerns with the statement of research questions on page 6:

1) What is the nature and extent to which the peer-reviewed, suicide loss and bereavement literature addresses engagement in meaningful activities of everyday living (i.e., occupation)?

2) What key facilitators and/or barriers impacting engagement in meaningful activities post-loss are described within the literature?

The above two broadly stated research questions do not address aspects stated in the abstract (Conclusions Part):

“Experiencing a suicide loss increases the risk of negative mental health outcomes. However, engagement in meaningful activities may improve mental health outcomes and have implications for improving supports and services for those bereaved.”

It seems the proposal needs to consider also reviewing—what type of mental health concerns are reported by people who have been exposed to and affected by suicide, and suicide bereaved short and long term (as stated on page 7). The question of whether exposure, etc. increases the risk of mental health issues is not addressed in the questions. This would require reviewing studies that specifically examine those studies assessed risks associated with exposure etc. Furthermore, it would be helpful to define these terms: “exposed to” “affected by” and “bereaved short” and long term.” It would also be helpful to define “meaningful activities” – are these specifically undertaken for coping purposes (e.g., seeing a therapist, joining a self-help group, reading self-help books on bereavement)? Can continuing to engage in daily living activities (e.g., brushing one’s teeth, taking showers, eating meals) be meaningful? It is not clear whether they plan to examine if they plan on reviewing whether engaging in meaningful activities indeed improves mental health outcomes. Will they be identifying the “meaningful” activities that studies have found to improve mental health outcomes? It would be helpful to have these points clarified in the proposal.

Reviewer #2: Overall, this is a solid manuscript, what's missing are the results and the interpretation, as well as some minor edits throughout the manuscript.

For the intro:

- be more specific, add in the stats and number detailing the decline of mental health status

- for the term "suicide survivor" - This label may be misconstrued (e.g., readers may interpret this as someone who survived a suicide attempt), add additional context or reframe.

- Re: Cerel et al reference (lines 33 through 35) - Add in specific reference for Canada. If there are limited studies or no studies, mention this. By doing so, it provides context as to why the Cerel et al reference was used.

- Re: Cerel et al reference (lines 37 - 39): Additional References? Supplement and support Cerel at al reference with additional references/studies.

- Awkward sentence: (Lines 50 - 54): Awkward sentence; rephrase this sentence - make it more concise.

For the Methods:

- Brief explanation as to why you chose the scoping review protocol by Arksey & O'Malley

- Brief explanation of whether or not the "two reviewers" are the same throughout this process or will they be rotated.

- Table 1  Population - Consider adding demographics and other details/information about the makeup of this population

- line 125 - grammar  ...began to develop [to] what it is today...  change to  "into"; rework this sentence, sounds awkward.

- Exclusion Criteria - Consider adding in a brief explanation as to why these are not being included.

- Stage 3 - study selection - Provide a brief explanation of how this will be randomized.

- Quality Appraisal - Provide 1-2 examples

For the Limitations

- When results are ready - describe the results, interpret the results, discuss the potential implications and significance of the study.

- Discuss/identify next steps. Add in discussion/conclusion

7. PLOS authors have the option to publish the peer review history of their article (what does this mean?). If published, this will include your full peer review and any attached files.

Reviewer #1: No

Reviewer #2: No

====

  Grant J. Rich, PhD   Twitter/X: @GrantJRich4APA 2023 Frances Mullen Award for Distinguished Contributions to International Psychology     (ICP: International Council of Psychologists) President-Elect Society for Peace, Conflict, and Violence (APA) President-Elect Society for Media Psychology and Technology (APA)  Fellow, Association for Psychological Science (APS) Fellow, American Psychological Association (APA) (D1, D2, D46, D48, D52)  Senior Contributing Faculty, Walden University, Juneau, Alaska Dr. Rich's SPN Website: http://rich.socialpsychology.org/ **Editorial Board Member:**
*PLOS ONE*, APA's *Peace & Conflict*, APA's *Traumatology*
 **Book Series Co-Editor** w/Anthony Marsella (U. Hawai'i) Springer. International & Cultural Psychology (ICUP)      https://www.springer.com/series/6089 **Select Recent Books** (Rich, Gielen, & Takooshian, 2017).* Internationalizing the Teaching of Psychology.* IAP.  (Rich & Sirikantraporn, 2018). *Human Strengths and Resilience: Cross Cultural and International Perspectives*. Rowman & Littlefield.  (Rich, Jaafar, & Barron, 2020).* Psychology in Southeast Asia*. Routledge. (Rich & Ramkumar, 2022).* Psychology in Oceania and the Caribbean*. (Foreword by past APA President Frank Worrell). Springer.   (Rich, Kuriansky, Gielen, & Kaplan, 2023). *Psychosocial Experiences and Adjustment of Migrants: Coming to the USA*.      (Foreword by past APA President Tony Puente).  Elsevier. (Rich, Kumar, & Farley, in contract). *Handbook of Media Psychology and Technology-The Science and the  Practice*. Springer.   

---

## [Author Response · Author response to Decision Letter 0]

1 Dec 2023

Dear Dr. Grant Rich, Academic Editor, 

Thank you for the review of our manuscript, entitled "Engagement in meaningful activities post suicide loss: A scoping review protocol" for publication in PLOS One and for all the helpful comments. We are pleased to submit an updated manuscript with tracked changes. 

Below, please find responses to each of the editor’s / reviewers’ comments and our response. 

1) Editor Comments and Responses

Issue raised: Please ensure that your manuscript meets PLOS ONE's style requirements, including those for file naming. 

Response: The formatting of the manuscript has been changed, throughout, to meet PLOS ONE style requirements.

Specific change: Changes to formatting of the manuscript: title page, headings, and body. 

Page #: 1-20

Issue raised: Please amend either the abstract on the online submission form (via Edit Submission) or the abstract in the manuscript so that they are identical.

Response: Apologies for this omission, we have now ensured that the abstract submitted online matches the abstract included within the manuscript.

Specific change: Abstract submitted online has been changed to match this manuscript

Page #: 2

Issue raised: We notice that your supplementary file is included in the manuscript file. Please remove them and upload them with the file type 'Supporting Information'. Please ensure that each Supporting Information file has a legend listed in the manuscript after the references list.

Response: The supplementary file has been removed from the manuscript file and will be attached separately. As well, a supporting information legend has been listed in the manuscript after the references list.

Specific change: Supplementary files added to submission and supporting information legend added to manuscript.

“S1 Appendix A. This is the Search Strategy for MEDLINE via Ovid. S2 Appendix B. This is the PRISMA-P 2015 Checklist”

Page #: 20

Issue raised: Please review your reference list to ensure that it is complete and correct. If you have cited papers that have been retracted, please include the rationale for doing so in the manuscript text, or remove these references and replace them with relevant current references. Any changes to the reference list should be mentioned in the rebuttal letter that accompanies your revised manuscript. If you need to cite a retracted article, indicate the article’s retracted status in the References list and also include a citation and full reference for the retraction notice.

Response: We have reviewed the reference list carefully. No retracted articles are included within this list. Slight changes to formatting of two reference were made.

Specific change: “JBI Critical Appraisal Tools. [cited 28 Jul 2023]. Available: https://jbi.global/critical-appraisal-tools”

“Cain AC. Survivors of suicide. Springfield, Ill: Thomas; 1972.”

Page #: 17-20

2) Reviewer 1 Comments and Responses

Issue raised: The proposal proposes to review studies (published after 1969) on post-loss engagement in meaningful activities by those people who were affected by suicide. The proposal has been pre-registered on Open Science Framework Registries. The proposal follows relevant guidelines for conducting scoping reviews. I recommend its publication if the journal has a policy to publish study proposals rather than completed studies.

Response: Thank you for your feedback. We have responded to each of your comments below. 

Specific change: N/A

Page #: N/A

Issue raised: I do have some concerns with the statement of research questions on page 6:

1) What is the nature and extent to which the peer-reviewed, suicide loss and bereavement literature addresses engagement in meaningful activities of everyday living (i.e., occupation)?

2) What key facilitators and/or barriers impacting engagement in meaningful activities post-loss are described within the literature?

The above two broadly stated research questions do not address aspects stated in the abstract (Conclusions Part):

“Experiencing a suicide loss increases the risk of negative mental health outcomes. However, engagement in meaningful activities may improve mental health outcomes and have implications for improving supports and services for those bereaved.”

Response: Thank you for this observation. The abstract has been revised to better align with the broad research questions of this scoping review. The aim of this scoping review is to provide a map of available peer-reviewed literature as it relates to suicide loss and engagement in meaningful activity, rather than reviewing literature to determine whether engaging in meaningful activity improves mental health outcomes.

Specific change: “Conclusion: Experiencing a suicide loss increases the risk of negative mental health outcomes. A synthesis of literature is required to map the current available evidence related to suicide bereavement and engagement in meaningful activities, with potential implications for improving supports and services for those bereaved.”

Page #: 1

Issue raised: It seems the proposal needs to consider also reviewing—what type of mental health concerns are reported by people who have been exposed to and affected by suicide, and suicide bereaved short and long term (as stated on page 7). The question of whether exposure, etc. increases the risk of mental health issues is not addressed in the questions. This would require reviewing studies that specifically examine those studies assessed risks associated with exposure etc.

Response: The following changes were made:

1) Clarification in the manuscript that mental health concerns have not been explored in literature using Cerel and colleagues proposed continuum of survivorship

2) Clarification that it is well documented through research that experiencing a suicide loss can pose a risk to the mental health of those impacted. Further addition of citations regarding the research evidence on mental health concerns seen across the continuum (without categorization)

3) Clarification on the gap being filled that better leads the reader into the purpose of this review: to gain an understanding of the literature landscape to synthesize findings, rather than exploring research related to a specific outcome (e.g., whether exposure to suicide loss increases the risk of mental health issues or whether engaging in meaningful activity improves mental health outcomes)

Specific change: 1) “Research has yet to be completed on specific physical and mental health consequences as they relate to the categories outlined in the proposed continuum of survivorship. [8] However, research conducted with survivors of suicide loss across the continuum suggests that they are at a higher risk of experiencing negative physical and mental health outcomes as compared to the general population. [2,4,11,12]”

2) “An exposure to suicide has also shown to increase the risk of post-traumatic stress disorder [13], depression[14], complicated grief [15], and suicidal ideations [16]. Further, Linde and colleagues studied the grief process following the experience of suicide loss, concluding that it is highly complex due to factors impacting those left behind.[17] In addition to the bereavement of an individual, survivors of suicide loss experience stigma, shame, guilt, need to conceal events surrounding death, blame, rejection, exposure to prior suicide attempts by deceased, familial history of mental health challenges, and strained relationships prior to and following the death.[17]” 

3) “To date, a review synthesizing the literature as it relates to engagement in meaningful activities post suicide loss has yet to be completed.”

Page #: 4-5, 6

Issue raised: Furthermore, it would be helpful to define these terms: “exposed to” “affected by” and “bereaved short” and long term.” It would also be helpful to define “meaningful activities” – are these specifically undertaken for coping purposes (e.g., seeing a therapist, joining a self-help group, reading self-help books on bereavement)? Can continuing to engage in daily living activities (e.g., brushing one’s teeth, taking showers, eating meals) be meaningful?

Response: The following changes were made: 

1) Definitions for the terms related to the continuum of survivorship by Cerel and colleagues in 2014, specifically: “exposed to”, “affected by”, “bereaved short term” and “bereaved long term”, have been added to the manuscript. 

2) As well, a definition of meaningful activity has been expanded within the manuscript. 

Specific change:

1) “Cerel and colleagues suggest a continuum of survivorship with the following four categories (from low to high degree of closeness): 1) suicide exposed (e.g., those who know of someone who died by suicide; however, did not personally experience psychological distress / consequences related to the loss or any long term impacts) ; 2) suicide affected (e.g.,, those who lost someone to suicide and experienced psychological distress) ; 3) suicide bereaved, short term (e.g., those who were close to / had an attachment relationship to the individual who died by suicide and experienced a short term response to bereavement); and 4) suicide bereaved, long term (e.g., those who were close to / had an attachment relationship to the individual who died by suicide and experienced a long term response to bereavement).[8]”

2) “As described within occupational therapy literature, this scoping review will define meaningful activities as any task related to self-care, productivity, and leisure “that is performed with some consistency and regularity, that brings structure, and is given value and meaning by individuals and a culture” (i.e., occupation).[21,22 p. 19]”

Page #: 4-5

Issue raised: It is not clear whether they plan to examine if they plan on reviewing whether engaging in meaningful activities indeed improves mental health outcomes. Will they be identifying the “meaningful” activities that studies have found to improve mental health outcomes? It would be helpful to have these points clarified in the proposal.

Response: Changes have been made throughout the manuscript (including within the abstract and introduction) to further clarify the purpose of this scoping review and re-emphasize the gap addressed by this review.

Specific change: 

“A synthesis of literature is required to map the current available evidence related to suicide bereavement and engagement in meaningful activities, with potential implications for improving supports and services for those bereaved.”

“Due to complexities related to the experience of suicide loss, the negative mental and physical health outcomes, and the increased risk of experiencing complicated grief, it is imperative that further research be undertaken. This research must focus on exploring concepts such as engagement in meaningful activities, that have shown potential in other contexts to act as protective factors to assist in bettering supports and services for those experiencing a loss. To date, a review synthesizing the literature as it relates to engagement in meaningful activities post suicide loss has yet to be completed. The objectives of this proposed scoping review are to: 1) explore the nature and extent to which the peer-reviewed, suicide loss and bereavement literature addresses engagement in meaningful activities of everyday living; and 2) identify key facilitators and/or barriers that may impact engagement in meaningful activities post-loss.”

Page #: 2, 6

2) Reviewer 2 Comments and Responses

Issue raised: Overall, this is a solid manuscript, what's missing are the results and the interpretation, as well as some minor edits throughout the manuscript.

Response: Thank you for your feedback. We have responded to each of your comments below. 

Specific change: N/A

Page #: N/A

Issue raised: For the intro:

1) Be more specific, add in the stats and number detailing the decline of mental health status

2) For the term "suicide survivor" - This label may be misconstrued (e.g., readers may interpret this as someone who survived a suicide attempt), add additional context or reframe.

3) Re: Cerel et al reference (lines 33 through 35) - Add in specific reference for Canada. If there are limited studies or no studies, mention this. By doing so, it provides context as to why the Cerel et al reference was used.

4) Re: Cerel et al reference (lines 37 - 39): Additional References? Supplement and support Cerel et al reference with additional references/studies.

5) Awkward sentence: (Lines 50 - 54): Awkward sentence; rephrase this sentence - make it more concise.

Response: 

1) We have added statistics (percentages) from Statistics Canada to better describe the decline in mental health status among Canadians.

2) The definition of survivors of suicide loss, suicide survivors or individuals bereaved by suicide is provided within the first paragraph of the manuscript. It is noted that survivors of suicide loss will be used within this paper. Further clarification has been added by describing these individuals as “those left behind” following a suicide. 

3) Clarification has been provided as to why this statistic was added within the manuscript providing better context for the reader. 

4) Additional references have been added to provide a fulsome understanding of the literature related to categorization of survivors of suicide loss, and the current state of the literature as it relates to the definition of survivors of suicide loss. 

5) This sentence has been revised and rephrased to ensure clarity.

Specific change: 

1) “According to Statistics Canada, 77% of youth and young adults ages 15-30, 75% of adults ages 31-46, and 72% of middle aged to older adults ages 47 and older, reported having excellent to good mental health in 2007.[1] This percentage has steadily declined to 60% among youth and young adults, 66% among adults, and 69% among middle aged to older adults in 2019.[1]”

2) “Each suicide leaves behind family members, friends, peers, and entire communities to manage the traumatic aftermath of this sudden loss. Those left behind are often referred to as survivors of suicide loss, suicide survivors or individuals bereaved by suicide.[4] These individuals will be referred to as survivors of suicide loss throughout this scoping review protocol.”

3) “Each suicide death leaves behind approximately six individuals.[5] This estimate was not based on empirical evidence, however, has been echoed widely.[6] This estimate closely aligns with what has been reported through Statistics Canada with estimates of 7 to 10 individuals impacted by each suicide loss.[2] However, more recently a random-digit dial survey found that approximately 135 individuals are exposed to each suicide within the United States.[6] This large discrepancy questions this widely accepted estimate and further speaks to the importance of understanding the impact of loss on the lives of those exposed to suicide.”

4) “Although consensus among researchers has yet to be reached regarding the definition of survivors of suicide loss, recent studies are moving towards providing a common language to identify and categorize the relationship and self-reported degree of closeness to the deceased.[7–10]”

5) “Further, Linde and colleagues studied the grief process following the experience of suicide loss, concluding that it is highly complex due to factors impacting those left behind.[17] In addition to the bereavement of an individual, survivors of suicide loss experience stigma, shame, guilt, need to conceal events surrounding death, blame, rejection, exposure to prior suicide attempts by deceased, familial history of mental health challenges, and strained relationships prior to and following the death.[17]”

Page #: 3-5

Issue raised: For the Methods:

1) Brief explanation as to why you chose the scoping review protocol by Arksey & O'Malley

2) Brief explanation of whether or not the "two reviewers" are the same throughout this process or will they be rotated.

3) Table 1: Population - Consider adding demographics and other details/information about the makeup of this population

4) Line 125 – grammar: ...began to develop [to] what it is today...  change to  "into"; rework this sentence, sounds awkward.

5) Exclusion Criteria - Consider adding in a brief explanation as to why these are not being included.

6) Stage 3 - study selection - Provide a brief explanation of how this will be randomized.

7) Quality Appraisal - Provide 1-2 examples

Response: Please see below for all changes made according to each point listed:

1) A brief description has been added to the manuscript regarding the choice of a mixed methodological approach (Arksey and O’Malley, Levac and colleagues, and the JBI methodology). 

2) This has been clarified in the manuscript. Two independent reviewers will screen each article through the title and abstract review, and full text review. However, reviewers will not be the same for each article and multiple reviewers will be involved throughout the process.

3) Population demographics will be explored through examination of descriptive statistics of included articles. Aspects of sample descriptions to be explored have been added to the manuscript. 

4) Grammar was corrected, and sentence revised / clarified. 

5) A brief explanation of reasoning for exclusion criteria is included within the body of the manuscript.

6) Method used to select articles for the pilot has been clarified in the manuscript.

7) An example has been provided to illustrate how critical appraisal tools will be chosen to evaluate the quality of each included article.

Specific change:

1) “This mixed methodological approach was chosen as authors were familiar with Arksey and O’Malley’s review stages, however, recognized that updates from Levac and colleagues, as well as the JBI methodology are essential as they increase rigour and clarity.[29–31] Within each stage outlined below, Levac and colleagues methodology, as well as the JBI methodology is reviewed to ensure all steps are rigorous.[30,31] For example, within stage 1, the JBI methodology suggests defining and aligning the objectives of the scoping review to the research questions which is completed below.”

2) “Reviewers will then consult the research team to discuss conflicts or possible areas of improvement of the study selection process. Two independent reviewers will screen each article through the title and abstract review, and full text review. Reviewers will not be the same for each article and multiple reviewers will be involved throughout the process.”

3) “For all studies the following descriptive information (i.e., data items) will be collected: 1) title of the study; 2) author(s); 3) name of journal; 4) publication year and country; 5) specific study design; and 6) sample description (e.g., age, sex, gender, time since loss, and nature of relationship with individual lost) and size.”

4) “Research on suicide postvention and post-loss intervention gained momentum in the late 1960’s.[33,34] During this time, terminology concerning the concept of suicide loss began to develop into what it is today.[8,33,34]”

5) “In addition to barriers such as time and resource constraints, these criteria will enable the exclusion of non-peer reviewed work which will assist in ensuring studies included within the review will be of better methodological quality.”

6) “The first 25 articles to appear on Covidence will be selected.[31]”

7) “For example, if an included study uses a qualitative design, the checklist for qualitative design referenced by the JBI will be used to critically appraise the quality of the study.[39]”

Page #: 7-9, 11-13 

Issue raised: For the Limitations:

1) When results are ready - describe the results, interpret the results, discuss the potential implications and significance of the study. Discuss/identify next steps. Add in discussion/conclusion.

Response: Thank you for all your suggestions, please see below for response:

1) As this is a scoping review protocol, these sections will be included in the final manuscript following the completion of the review. Upon completion, results will be available that will help inform next steps, discussion, and the conclusion. This scoping review protocol used the framework provided by the Joanna Briggs Institute (JBI) with the aim of following a standardized method. Please refer to Chapter 11 of the JBI for more information: Peters MD, Godfrey C, McInerney P, Munn Z, Tricco AC, Khalil H. Chapter 11: Scoping Reviews. In: Aromataris E, Munn Z, editors. JBI Manual for Evidence Synthesis. 2020. Available: https://doi.org/10.46658/JBIMES-20-12

Specific change: No change made.

Page #: N/A

Thank you again to the editor and both reviewers for their commitment in reviewing this manuscript. We hope that the updated manuscript is seen as an improvement and that the responses above are clarifying. 

All authors continue to declare no conflict of interest with this research study or publication. 

Respectfully submitted,

Monique Gill, PhD Candidate, OT Reg. (Ont.), MScOT, BHSc

PhD Candidate, Rehabilitation Sciences Institute

University of Toronto

Email: moniquekaur.gill@mail.utoronto.ca

---

## [Editor Report · Decision Letter 1]

14 Dec 2023

Engagement in meaningful activities post suicide loss: A scoping review protocol

PONE-D-23-25565R1

Dear Monique and coauthors

We’re pleased to inform you that your manuscript has been judged scientifically suitable for publication and will be formally accepted for publication once it meets all outstanding technical requirements.

Kind regards,

Grant Rich, Ph.D.

Academic Editor

PLOS ONE

Additional Editor Comments (optional):

Thank you for making the minor revisions requested and I am happy to accept this article which makes a valuable contribution to psychology and related disciplines. Dr Rich

Reviewers' comments:

Grant J. Rich, PhD  Twitter/X: @GrantJRich4APA2023 Frances Mullen Award for Distinguished Contributions to International Psychology    (ICP: International Council of Psychologists)President-Elect Society for Peace, Conflict, and Violence (APA)President-Elect Society for Media Psychology and Technology (APA) Fellow, Association for Psychological Science (APS)Fellow, American Psychological Association (APA) (D1, D2, D46, D48, D52) Senior Contributing Faculty, Walden University, Juneau, AlaskaDr. Rich's SPN Website: http://rich.socialpsychology.org/**Editorial Board Member:**
*PLOS ONE*, APA's *Peace & Conflict*, APA's *Traumatology*
**Book Series Co-Editor** w/Anthony Marsella (U. Hawai'i) Springer. International & Cultural Psychology (ICUP)     https://www.springer.com/series/6089**Select Recent Books**(Rich, Gielen, & Takooshian, 2017).* Internationalizing the Teaching of Psychology.* IAP. (Rich & Sirikantraporn, 2018). *Human Strengths and Resilience: Cross Cultural and International Perspectives*. Rowman & Littlefield. (Rich, Jaafar, & Barron, 2020).* Psychology in Southeast Asia*. Routledge.(Rich & Ramkumar, 2022).* Psychology in Oceania and the Caribbean*. (Foreword by past APA President Frank Worrell). Springer.  (Rich, Kuriansky, Gielen, & Kaplan, 2023). *Psychosocial Experiences and Adjustment of Migrants: Coming to the USA*.     (Foreword by past APA President Tony Puente).  Elsevier.(Rich, Kumar, & Farley, in press). *Handbook of Media Psychology and Technology-The Science and the  Practice*. Springer.

---

## [Editor Report · Acceptance letter]

20 Dec 2023

PONE-D-23-25565R1 

PLOS ONE

Dear Dr. Gill, 

I'm pleased to inform you that your manuscript has been deemed suitable for publication in PLOS ONE. Congratulations! Your manuscript is now being handed over to our production team.

Kind regards, 

on behalf of

Dr. Grant Rich 

Academic Editor

PLOS ONE